# Multiple Adenocarcinomas of the Small Bowel in a Patient with Brunner's Glands Agenesia: A Previously Unreported Association

Sergio Coverlizza [1], Lavinia Masu [2] and Claudia Manini [1,3,*]

1   Department of Pathology, San Giovanni Bosco Hospital, 10154 Turin, Italy
2   Department of Pathology, S. Andrea Hospital, ASLVC, 13100 Vercelli, Italy
3   Department of Sciences of Public Health and Pediatrics, University of Turin, 10124 Turin, Italy
*   Correspondence: claudia.manini@aslcittaditorino.it

**Abstract:** Adenocarcinoma of the small bowel is rather uncommon and several etio-pathogenic factors have been proposed. We report a case of multiple synchronous adenocarcinomas arising in the non-ampullary duodenum and first tract of the jejunum in a background of Brunner's glands agenesia, chronic duodenitis, and extensive dysplasia in a 64 year-old woman. To the best of our knowledge such association has not been reported so far.

**Keywords:** Brunner's gland agenesia; intestinal dysplasia; small bowel adenocarcinoma; histology; pathological diagnosis

## 1. Introduction

Adenocarcinoma of the duodenum are the most frequent tumor in the small bowel [1]. Fifty to 70% of the cases arise in the duodenum. Although duodenum represents less than 3% of the total length of the small bowel, the relative high frequency of periampullary epithelial neoplasms might be explained by the local chemical injury caused by gastric acids and bile salts. Non-ampullary duodenal carcinomas are exceedingly rare [2]. Their incidence increases in cancer-predisposing syndromes like familiar adenomatous polyposis, and MUTYH-, NTHL1-, and PPAP-associated polyposis [3,4].

Brunner's glands are branched or coiled tubular glands located in the submucosa of the duodenum, particularly in the proximal segment, which produce mucinous secretions protecting the underlying epithelial cells from the chemical insult represented by the presence of gastric acid and bile in the area [1]. Protective substances present in their secretions include mucin glycoproteins, bicarbonate, epidermal growth factors, trefoil peptides, bactericidal factors, proteinase inhibitors, immunoglobulins, and surface-active lipids [1,5,6]. The second part of the duodenum at the level of the Vater's papilla and the third portion physiologically contain only scarce glands. Apart from that, Brunner's gland distribution is variable and may extend to the jejunum [7,8].

The present case associates Brunner's gland agenesia with multiple adenocarcinomas in the small bowel and leads us to hypothesize that the reduction, or absence, of the local Brunner's gland population might promote the development of chronic inflammation. This inflammation eventually could facilitate the development of malignancies. As far as we know, such association has never been reported.

## 2. Case Report

A 64-year-old woman presented with recurrent vomiting and remarkable weight loss in the past three weeks. A diagnosis of atrophic gastropathy and duodenal bulb ulcer was made one year before. She referred a history of hiatal hernia and several anemic episodes in the past years. Serological findings were under normal limits. Moreover, she was diagnosed and treated of breast carcinoma 15 years ago. Subsequent molecular studies confirmed that *BRCA1* and *BRCA2* genes were not mutated in the breast cancer tissue.

Physical exam revealed a moderately painful abdomen. Esophago-gastro-duodenoscopy displayed hiatal hernia with bile reflux and gastric atrophy without evident focal lesions. The second duodenal portion showed a partially stenosing mass that impeded the advance of the endoscope. Several biopsies were obtained from the duodenum showing severe dysplasia associated with intestinal-type adenocarcinoma. X-ray exam with oral barium and CT confirmed the presence of a tumor in the duodenum and a second tumor close to the duodenal-jejunal flexure. Predisposing factors of small bowel adenocarcinoma such as Crohn's disease, Lynch syndrome, Peutz-Jeghers syndrome, and celiac disease, were excluded. Colonoscopy was also performed showing only non-specific inflammatory changes. The patient underwent a wide surgical resection with Child reconstruction. The immediate post-surgical period was uneventful.

The patient received adjuvant chemotherapy with 5 cycles of etoposide-leucovorin-5-fluorouracil (ELF) and had been disease-free for 27 months after surgery. From there on, she developed mediastinal lymphadenopathies whose biopsy revealed a G3 adenocarcinoma attributable to the primary duodenal tumor. A metastatis from the breast cancer was ruled out. After that, the patient pursued a brisk physical decline and finally died.

### 3. Pathological Findings

The surgical specimen consisted of the pancreatic head, gastric antrum (6 cm. long), duodenum (16 cm. long) and jejunum (10 cm. long). Grossly, four masses were found in the intestinal wall classified as polypoid/sessile broad-based neoplasia (type Is) and superficial/depressed and elevated (type IIa+c) lesions according to Paris Classification of Early Cancer. The tumor located in the first jejunal loop was macroscopically an invasive neoplasm.

Histologically, the gastric antrum showed atrophic gastritis, multiple hyperplastic polyps and diffuse foveolar hyperplasia. The duodenum displayed extensive gastric pyloric metaplasia, atrophy, and multifocal dysplasia of varied degrees (categories 3 and 4 according to Vienna Classification). H.pylori was not detected. No Brunner's glands were found in the specimen after an exhaustive sampling (62 samples in 31 paraffin blocks) (Figure 1). Up to 4 different and distant one each other non-ampullary adenocarcinomas were found along the surgical specimen (Table 1) (Figure 2). In brief, tumors located the second and fourth part of the duodenum, and in the jejunum, were confined to the lamina propria and showed a low-grade well-differentiated histology, with well-formed glands and mild cytologic atypia. By contrast, the tumor located in the third portion of the duodenum was fully invasive across the intestinal wall and displayed a non-cohesive high-grade poorly-differentiated architecture with signet-ring cell morphology. A posterior pancreato-duodenal lymph node showed metastatic seed of signet-ring cell adenocarcinoma. The Vater's ampulla did not show pathological findings.

**Table 1.** Tumor location and main histological findings.

|  | Location | Histology | Grade | Stage |
|---|---|---|---|---|
| Tumor 1 | Duodenum (2nd part) | Well-diff. adenocarcinoma | 1 | pT1 |
| Tumor 2 | Duodenum (3rd part) | Signet-ring adenocarcinoma | 3 | pT4 |
| Tumor 3 | Duodenum (4th part) | Well-diff. adenocarcinoma | 1 | pT1 |
| Tumor 4 | Jejunum (1st loop) | Well-diff. adenocarcinoma | 1 | pT1 |

Mismatch repair gene immunohistochemical analysis was performed in the tumor with signet-ring cell features (tumor 2 in the Table) showing preservation of the nuclear staining with MLH-1, MSH-2, MSH-6, and PMS2 antibodies. Microsatellite instability

analysis (PCR) was not performed. No germline mutations in APC and MUTYH genes were found.

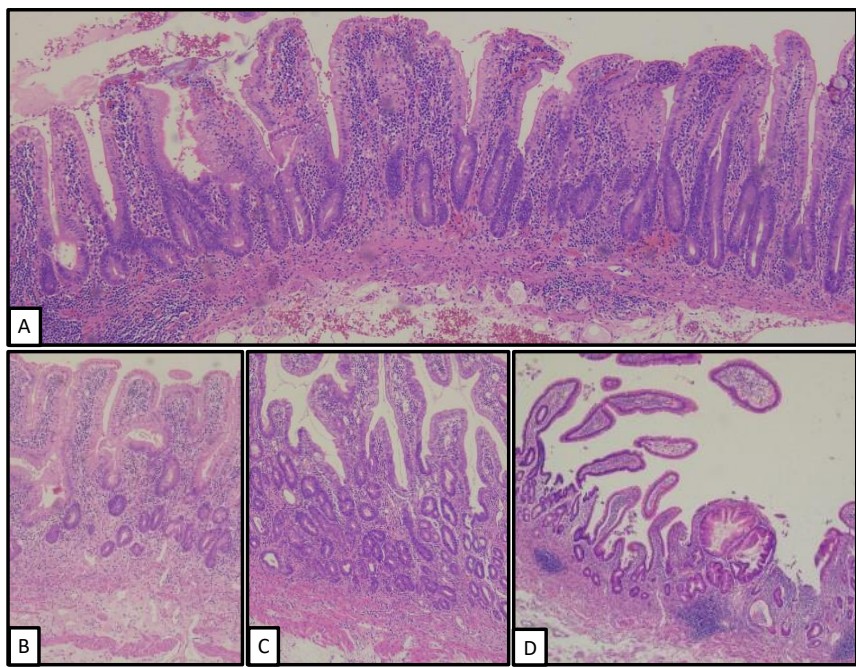

**Figure 1.** Low-power view of several portions of the small bowel wall showing Brunner's gland agenesia (**A**), chronic inflammation (**B**), low-grade (**C**), and high-grade (**D**) epithelial dysplasia.

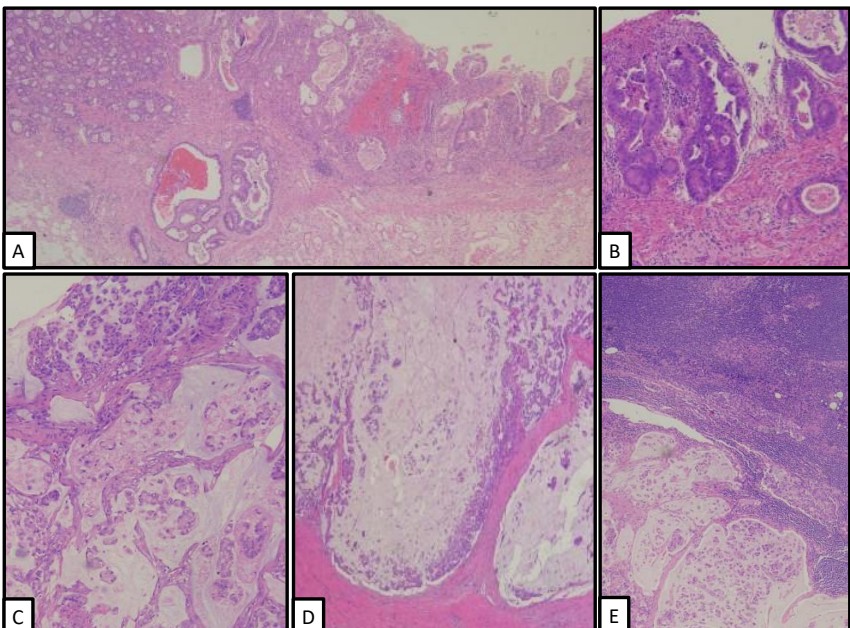

**Figure 2.** Low-power view of small bowel adenocarcinomas associated to Brunner's gland agenesia. Both, superficial low-grade adenocarcinoma (**A**,**B**), and invasive and lymph node metastasizing high-grade adenocarcinoma with signet-ring cell histology (**C–E**).

## 4. Discussion

Adenocarcinoma of the duodenum is the most frequent malignancy in the small bowel [1], but it is relatively uncommon if compared to colonic adenocarcinomas. Several factors have been proposed to explain the different risk of malignancy in the small and large bowel. Among them, a faster transit, the liquid content with dilution of luminal carcinogens,

a less mechanical stress, the protection offered by the mucosa-associated lymphoid tissue, and the action of the mucosal enzymes able to detoxify the luminal content are reasons argued in favor of the small bowel [6].

Several predisposing factors influence the development of severe epithelial dysplasia and adenocarcinomas in the small bowel: smoking, previous history of colorectal cancer, H. pylori infection [9], celiac disease [10], and familial adenomatous polyposis and other cancer-predisposing syndromes such as Lynch syndrome, Peutz-Jeghers syndrome, and others [4,11,12]. Moreover, chronic inflammation is a well-established risk factor associated to carcinogenesis in many organs, including the gastrointestinal tract [13,14]. Our patient had atrophic gastritis and active duodenitis. This fact may justify the extensive gastric metaplasia in the duodenum detected under the microscope, a condition that is usually associated with hyperplasia of the Brunner's glands. Although the absence of Paneth cells, goblet cells, and endocrine cells in the duodenal villi and crypts has been occasionally reported [15], the Brunner's gland agenesia seen in our patient has never been reported.

It is thought that Brunner's glands originate from the inward migration of stem cells through buds growing out of the duodenal crypt which subsequently acquire tubular or tubule-alveolar architecture in the submucosa same as neck cells migration and subsequent transformation into zymogenic cells in the gastric mucosa [16,17]. In particular, the role of trefoil peptides (proteins of the gastrointestinal tract regulating epithelial cell differentiation and mucus stabilization) in the development and function of the adult tubuloalveolar patter of Brunner's glands has been extensively investigated by Wright in a model mimicking their histogenesis: the ulcer-associated cell lineage [5]. In this model, trefoil peptides have been shown to be largely expressed, thus contributing to the mucosal defense even modifying gene expression in indigenous cell lineages of the surrounding mucosa. Once more, it should be taken into account that Brunner's glands secretion protects the underlying epithelial cells from the chemical insult represented by the presence of gastric acid and bile. Aside from trefoil peptides, such protection is made also by mucin glycoprotein, bicarbonate, epidermal growth factors, bactericidal factors, proteinase inhibitors, and surface-active lipids [5,6].

In the light of these data, it seems conceivable to hypothesize that the complete absence of Brunner's glands in our patient could predispose to the development of neoplasia by depriving the small intestine mucosa of the physiological protective factors. The clinical history of the patient here presented shows she had been suffering of gastrointestinal symptoms for years, and the colonic biopsies performed displayed features consistent with inflammatory bowel disease. However, a diagnosis of Crohn's disease was never made. Brunner's gland agenesia in our patient was associated with extensive dysplasia which ultimately led to the development of multiple adenocarcinomas. The occurrence of 4 independent tumors in the patient rises the possibility of a local metastatic seed from one of them. However, this seems improbable because three of them were low-grade and the other, that was high-grade, displayed an utterly different histology. At least theoretically, the case here presented shows that Brunner's gland agenesia could be considered as an unusual precancerous condition. No other predisposing factors were detected.

The prognosis of duodenal adenocarcinoma is largely dependent on the pathological stage. Radical surgery is feasible in 50% of the cases and represents the first objective of treatment [18]. As for the colorectal cancer, lymph node involvement is the single most important prognostic factor, to the point that 5-year survival rates vary widely according to the nodal status. Adjuvant chemotherapy is frequently administered, especially in the lymph node positive cases. However, a precise therapeutic strategy has not been detailed so far due to the relative rarity of duodenal cancers [19]. In our case, one of the four adenocarcinomas was high-grade, pT4, and lymph node positive. This fact led to the patient's death despite chemotherapy.

## 5. Conclusions

The case here presented represents the first documented case of Brunner's gland agenesis associated with multiple adenocarcinomas of the small bowel. Based on this case, we hypothesize that the absence of Brunner's gland may be a risk factor for developing malignancies at this level.

**Author Contributions:** S.C., L.M. and C.M. analyzed and diagnosed the case; S.C. and L.M. reviewed the bibliography; C.M. wrote the paper. All authors have read and agreed to the published version of the manuscript.

**Funding:** This research received no external funding.

**Institutional Review Board Statement:** Not applicable.

**Informed Consent Statement:** Patient consent was waived due to patient's death.

**Conflicts of Interest:** The authors declare no conflict of interest.

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
