# Peer review of "Multiple Adenocarcinomas of the Small Bowel in a Patient with Brunner’s Glands Agenesia: A Previously Unreported Association"

_clinpract, doi:10.3390/clinpract12050069_

Round 1

Reviewer 1 Report

The authors reported a case with multiple synchronous adenocarcinoma in non-ampullary duodenum and jejunum with background of brunnder's gland agenesia. Brunner's gland can secrect alkaline secretion to protect the duodenum from acidic content of chyme. In hypothesis, brunner's gland can agenesis can be related to duodenal inflammation and duodenitis.

The finding was interesting and may lead to further investigation of pathogenesis of duodenal adenocrcinoma. There are several issues may be condsidered to be further discussed.

1. The patient had history of breast carcinoma, did she had BRCA1 or BRCA2  mutation?. Is her breast cancer related to brunnner's gland agenesia , duodenal 

carcinoma or both?

2. The ptient had histiry of atrophic gastropathy, did she had h.pylori infection history or treated before? Or she had history of frequent NSAIDs use?

3.It will be great if there's endoscopy picture and gross specimen photo for better description of the patient. 

4. Is there any way to detect brunner's gland agenesis clincally? by intestinal fluid acquisition? endoscopy evaluation ? confocal endomicroscopy? or endoscopic histologic  examionation? This could be meaningful to early dectect potential risky patient for further duodenal adenocarcinoma occurence.

Author Response

We are thankful for the comments which are absolutely pertinent. Due to the patient died long time ago the accession to further information is limited. Some of them, however, will be added in the clinical case report.

  1. The patient had history of breast carcinoma, did she had BRCA1 or BRCA2  mutation?. Is her breast cancer related to brunnner's gland agenesia , duodenal carcinoma or both?

The breast cancer was not mutated. This information has been added in the case report.

An association between breast cancer, Brunner gland agenesia, and small bowel carcinoma in the patient can be considered as a possibility, but we have not found any previous case showing such association so far. Consequently, we have been cautious in our description of an association never reported before. Further examples of such association will give interesting clues to answer this question.

  1. The ptient had histiry of atrophic gastropathy, did she had h.pylori infection history or treated before? Or she had history of frequent NSAIDs use?

Gastric atrophy in our patient included intestinal metaplasia, so H. Pylori was not detected. No previous history of this infection neither frequent NSAID use was found on the history records.  

  1. It will be great if there's endoscopy picture and gross specimen photo for better description of the patient.

Yes, we fully agree. It would be great to have endoscopic and/or macroscopic pictures of the case. Regrettably, we do not have them. 

  1. Is there any way to detect brunner's gland agenesis clincally? by intestinal fluid acquisition? endoscopy evaluation ? confocal endomicroscopy? or endoscopic histologic  examionation? This could be meaningful to early dectect potential risky patient for further duodenal adenocarcinoma occurence.

We agree with the reviewer’s suggestions. Al least theoretically, all these possibilities would add further information to make the diagnosis. However, Brunner´s gland agenesia is so rare that, in practical terms, the efforts proposed by the reviewer are not available in the routine practice of public health systems. We agree that endoscopic biopsies could alert about this possibility. Endoscopic biopsies are performed routinely in our context. For this reason, this paper may be useful. Pathologists analyzing such biopsies must be aware of this possibility.

Reviewer 2 Report

Dear Authors,

Congratulations!

But you could ask for more imagistics like a ct scan or the X-ray exam with oral barium for a wider audience in the future for this paper!

Author Response

Thank you very much for your comments. Yes, CT scan and/or X-ray exam, and even a gross microscopic picture, would have been of much help. Unfortunately, we do not have them. The case, however, is useful in the sense that it makes the pathologists be aware of this absolutely exceptional condition. This is the message we want to transmit.

Reviewer 3 Report

Dear Authors,

This study intriguingly implies a novel association between Brunner’s gland agenesia and carcinogenesis of the small bowel. I think several points should be clarified before publication.

 First, known predispositions of small bowel adenocarcinoma (SBA) have to be excluded, such as Crohn’s disease, FAP, Lynch syndrome, Peutz-Jeghers syndrome, and celiac disease. Symptoms (chronic diarrhea), endoscopic findings (polyposis, inflammatory findings), and serological findings (EMA-IgA, tTG-IgA) need to be more clearly explained. In the page3 line 95 to 97, I’m afraid microsatellite instability (MSI) analysis and mismatch repair (MMR) protein expression analysis were confused. Tumor DNA is extracted from tumor tissue and is analyzed with polymerase chain reaction (PCR) in MSI analysis. The results include “MSI-high” and “MSI-low or MSS (microsatellite stable)”. Please describe the result. Immunohistochemical staining is performed on pathological specimen using anti-MLH-1, anti-MSH2, anti-MSH6 and usually anti-PMS2 antibody in MMR protein expression analysis. They are different. EPCAM protein is also known as one of MMR proteins. The reason why PMS2 and EPCAM were not analyzed is preferably clarified.

 Second, please explain that the four lesions were independent. There is a possibility that the small three lesions were metastases from the advanced one. It is unlikely, because their histological types were different. However, the possibility should be mentioned and denied in discussion.

 Third, were examinations thoroughly performed to confirm Brunner’s gland agenesia? Brunner’s gland agenesia is an uncommon condition without specific definition. Authors wrote that "an exhaustive sampling" was performed. Please describe it concretely. How many samplings were made to confirm “agenesia”. As to the endoscopic findings, gastric atrophy was revealed. How about the duodenal atrophy? Was H. pylori infection present?

 There are subtle grammatical mistakes. Please have the manuscript checked by a native speaker before submission.

Page 2 line 48, “before” should be “ago”.
 Page 2 line 50 “gastric atrophy gastric” should be “gastric atrophy”.
 Page 2 line 59 “fluorouracile” should be “fluorouracil”.
 Page 2 line 59-60 “followed a disease-free period of 27 months after surgery” should be “had been disease free for 27 months after surgery”
 Page 2 line 61, “A metastatic seed from the mammary gland” should be “A metastasis from the breast cancer”.
 Page 2 line 70 “on naked eye” should be “macroscopically”.
 Page 4 line 119, “districts” should be “organs”.
 Page 5 line 122, “associated to” should be “associated with”.
 Page 5 line 127 and 131, “tubule-alveolar” should be "tubuloalveolar".

Regards,

Author Response

The authors thank the reviewers comments and suggestion that will improve the level of this contribution. Changes are highlighted in red.

First, known predispositions of small bowel adenocarcinoma (SBA) have to be excluded, such as Crohn’s disease, FAP, Lynch syndrome, Peutz-Jeghers syndrome, and celiac disease. Symptoms (chronic diarrhea), endoscopic findings (polyposis, inflammatory findings), and serological findings (EMA-IgA, tTG-IgA) need to be more clearly explained. In the page3 line 95 to 97, I’m afraid microsatellite instability (MSI) analysis and mismatch repair (MMR) protein expression analysis were confused. Tumor DNA is extracted from tumor tissue and is analyzed with polymerase chain reaction (PCR) in MSI analysis. The results include “MSI-high” and “MSI-low or MSS (microsatellite stable)”. Please describe the result. Immunohistochemical staining is performed on pathological specimen using anti-MLH-1, anti-MSH2, anti-MSH6 and usually anti-PMS2 antibody in MMR protein expression analysis. They are different. EPCAM protein is also known as one of MMR proteins. The reason why PMS2 and EPCAM were not analyzed is preferably clarified.

All the suggestions have been included in the text. Yes, we have added this important information in the case report. Also, mismatch repair protein detection has been performed, now including PMS2. Although they partially surrogate microsatellite instability, the reviewers is totally right with this point, so the text has been changed accordingly. However, we are not able to detect EPCAM protein in our laboratory.

Second, please explain that the four lesions were independent. There is a possibility that the small three lesions were metastases from the advanced one. It is unlikely, because their histological types were different. However, the possibility should be mentioned and denied in discussion.

Yes, done in the discussion.

Third, were examinations thoroughly performed to confirm Brunner’s gland agenesia? Brunner’s gland agenesia is an uncommon condition without specific definition. Authors wrote that "an exhaustive sampling" was performed. Please describe it concretely. How many samplings were made to confirm “agenesia”. As to the endoscopic findings, gastric atrophy was revealed. How about the duodenal atrophy? Was H. pylori infection present?

Yes, we understand your question. A total of 31 blocks containing 62 fragments (two fragments per block) of the non-tumor areas were analyzed. This figure has been included in parenthesis together with exhaustively sampled assertion.

No, as requested by the reviewer 1, H. pylori infection was nor detected.

There are subtle grammatical mistakes. Please have the manuscript checked by a native speaker before submission

Grammatical mistakes have been all included in the text. We thank the reviewer also for that. The text has been reviewed.

Round 2

Reviewer 3 Report

Dear authors,

Thank you for revising the manuscript. I believe that this study gives readers insight about carcinogenesis of small bowel adenocarcinoma.

There were some minor mistakes as follows.

In page 3 line 82, “H. pylorii” should be “H. pylori”.

In page 5 line 158, “microsatellite instability” was not assessed in this study, according to the revised manuscript.

 Regards,

Author Response

The authors thank Reviewer 3 for their useful comments.

  1. Yes, H. pylori was mispelled and the correct form has been included in the text.
  2. Yes, microsatellite instability has been removed from line 158